# Accurate quantification of circular RNAs identifies extensive circular isoform switching events

Jinyang Zhang [1,2], Shuai Chen[1], Jingwen Yang[1] & Fangqing Zhao[1,2,3]*

Detection and quantification of circular RNAs (circRNAs) face several significant challenges, including high false discovery rate, uneven rRNA depletion and RNase R treatment efficiency, and underestimation of back-spliced junction reads. Here, we propose a novel algorithm, CIRIquant, for accurate circRNA quantification and differential expression analysis. By constructing pseudo-circular reference for re-alignment of RNA-seq reads and employing sophisticated statistical models to correct RNase R treatment biases, CIRIquant can provide more accurate expression values for circRNAs with significantly reduced false discovery rate. We further develop a one-stop differential expression analysis pipeline implementing two independent measures, which helps unveil the regulation of competitive splicing between circRNAs and their linear counterparts. We apply CIRIquant to RNA-seq datasets of hepatocellular carcinoma, and characterize two important groups of linear-circular switching and circular transcript usage switching events, which demonstrate the promising ability to explore extensive transcriptomic changes in liver tumorigenesis.

[1] Computational Genomics Lab, Beijing Institutes of Life Science, Chinese Academy of Sciences, 100101 Beijing, China. [2] University of Chinese Academy of Sciences, 100049 Beijing, China. [3] Center for Excellence in Animal Evolution and Genetics, Chinese Academy of Sciences, 650223 Kunming, China. *email: zhfq@biols.ac.cn

Circular RNA (circRNA) is a large class of RNA molecules that contain a covalent circular structure formed by non-canonical 3′ to 5′ end-joining event called back-splicing. Previous studies have shown that circRNAs are widely present and some are conserved in eukaryotic organisms, and have a relative low abundance compared with canonical linear mRNA transcripts[1]. Although the exact function of most circRNAs is still ambiguous, studies have shown that circRNAs may function as sponges to sequester miRNAs or RNA binding proteins (RBPs)[2–4]. More recently, several studies found that internal ribosome entry sites or N6-methyladenosine can promote the translation of circRNAs, which results in the biogenesis of circRNA-derived proteins[5–7]. Thus, circRNAs have great potentials to play important roles in cellular metabolic process, and the characterization and quantification of circRNAs from high-throughput RNA-seq data has become an emerging problem in circRNA studies.

A number of computational methods have been developed for characterizing circRNAs[8]. Most of these methods employ alignment based strategies to recognize back-spliced junction (BSJ) reads from circRNAs, and have limited sensitivity and notable false-positive rate in circRNA identification[9]. Recently, a model-based strategy is employed by Sailfish-cir[10], which uses a quasi-mapping method to acquire direct estimation of circular transcript expression. This statistical model depends on the unique sequence between circular and linear transcripts, which limits its ability on quantifying exonic circRNAs. Moreover, the ratio of BSJ reads to canonical linear reads at the junction, which represents the splicing preference in precursor mRNAs, is also an important factor in circRNA analysis. However, among all currently available circRNA detection tools, only CIRI2[11,12] can output the junction ratio directly. DCC[13] and Sailfish-cir estimate the expression values of both linear and circular transcripts, which can be used to calculate circRNA's junction ratio. Hence, a reliable computational tool is urgently needed for accurate quantification of circRNAs and their parental linear transcripts.

Differential expression analyses of circRNAs in different samples is a routine analysis in circRNA studies. Currently, simple statistical tests (e.g., $t$-test) or differential expression analysis pipelines designed for linear RNA transcripts (e.g., DESeq2[14]) are often used to evaluate the significance of differentially expressed circRNAs. Since most of circRNAs are expressed at extremely low levels, RNase R treatment are usually performed to enrich circRNAs. For studies using RiboMinus/RNase R-treated RNA-seq libraries, the variation of enrichment coefficient in the RNase R treatment step may result in biased estimation of circRNA expression levels. In addition, analysis of experimental replicates has shown the poor reproducibility of circRNA identification[15], which further indicates that accurate characterization and quantification of circular transcripts is crucial in circRNA studies.

To overcome these limitations, we propose CIRIquant for accurate quantification of both circRNAs and their parental transcripts and filtration of false positives BSJ reads. CIRIquant can employ currently widely used tools (e.g., CIRI2[11,12], CIRCexplorer2[16], find_circ[2], etc.) for circRNA identification, then generated pseudo-reference sequences for the identified circRNA transcripts to re-align putative BSJ reads. Using both alignment results of reads against the reference genome and pseudo circRNA transcripts, CIRIquant not only achieves more accurate and sensitive identification of BSJ reads, but also enables reliable quantification of junction ratio for circRNAs. Moreover, CIRIquant provides a convenient function for one-stop circRNA differential expression analysis. We apply CIRIquant to survey circRNA expression profiles between hepatocellular carcinoma (HCC) tumor samples and their adjacent normal tissues, which unveil extensive transcriptomic changes in liver tumorigenesis.

Additionally, we profile the switching events in circRNA junction ratio and circular transcript usage, and characterize these two groups of circRNAs with potential biological functions. We believe that CIRIquant, which provides an accurate and efficient quantification approach to characterize circRNAs and perform differential expression analysis after correcting experimental and computational biases, will greatly improve our understanding of circRNA diversities and functions.

## Results

**Challenges in quantifying the expression of circRNAs**. To rigorously evaluate the challenges in current quantification of circRNA expression, we collected 63 transcriptomic samples from six previous studies[16–22], including both RiboMinus and RiboMinus/RNase R RNA-seq libraries of four species (human, mouse, fly and roundworm, Supplementary Table 1). All these RNA-seq datasets were aligned to their reference genomes and the ribosomal RNA (rRNA) sequences using HISAT2[23] to assess the mapping rate and rRNA sequence fraction. As shown in Fig. 1a, RNA-seq datasets from four species showed an extraordinarily high variance in the efficiency of rRNA sequence depletion, which is largely due to the limited specificity and efficiency of current RiboMinus transcriptome isolation kit. For each sample, the raw reads were further aligned to the reference genome using the BWA-MEM algorithm[24], and then subjected to CIRI2 for circRNA detection and quantification. We chose CPM (counts per million mapped reads) to represent circRNA expression levels to remove the biases derived from different library insert size and sequencing depth. For 34 pairs of RiboMinus and RiboMinus/RNase R samples, the overall number of detected circRNAs ranged from several thousand to over 30 thousand, in which most of circRNAs detected in the RiboMinus samples can be validated in the RiboMinus/RNase R samples (Fig. 1b). Although the total number of identified circRNAs increased after RNase R enrichment, over 50–80% highly expressed circRNAs could be detected in both RiboMinus and RiboMinus/RNase R samples. Considering that RNase R treatment is widely used for circRNA enrichment, we compared expression levels of both gene and circRNA in RNase R-treated and untreated samples to investigate whether RNase R treatment can introduce bias in transcript quantification. As shown in Fig. 1b (right), gene expression levels decreased over 2-fold, which is in concordance with the degradation effect of linear RNAs in RNase R treatment. However, the relative expression value of circRNAs also showed a certain level of reduction. Considering that the RNase R treatment is expected to increase the saturation level of circRNA detection, the more circRNAs are identified, the more circular reads and the lower the relative expression value for each specific circRNA.

To further investigate the effect of RNase R treatment on circRNA quantification, we divided the detected circRNAs into two groups, highly expressed circRNAs and lowly expressed circRNAs, according to the ranking of their expression values. Then, we calculated the proportion of reads from the two groups of circRNAs after RNase R treatment. As shown in Fig. 1c and Supplementary Fig. 1, we observed an increased proportion of BSJ reads for highly expressed circRNAs after RNase R treatment, indicating that highly expressed circRNAs tended to be enriched in RiboMinus/RNase R-treated data. In order to assess the efficiency of RNase R treatment, we calculated the enrichment coefficient for circRNAs in different RNA-seq datasets. Surprisingly, the enrichment coefficient exhibited distinct difference among samples from different species or even within the same species (Fig. 1d). Datasets from human samples showed the highest effectiveness of RNase R enrichment of circRNAs, with a

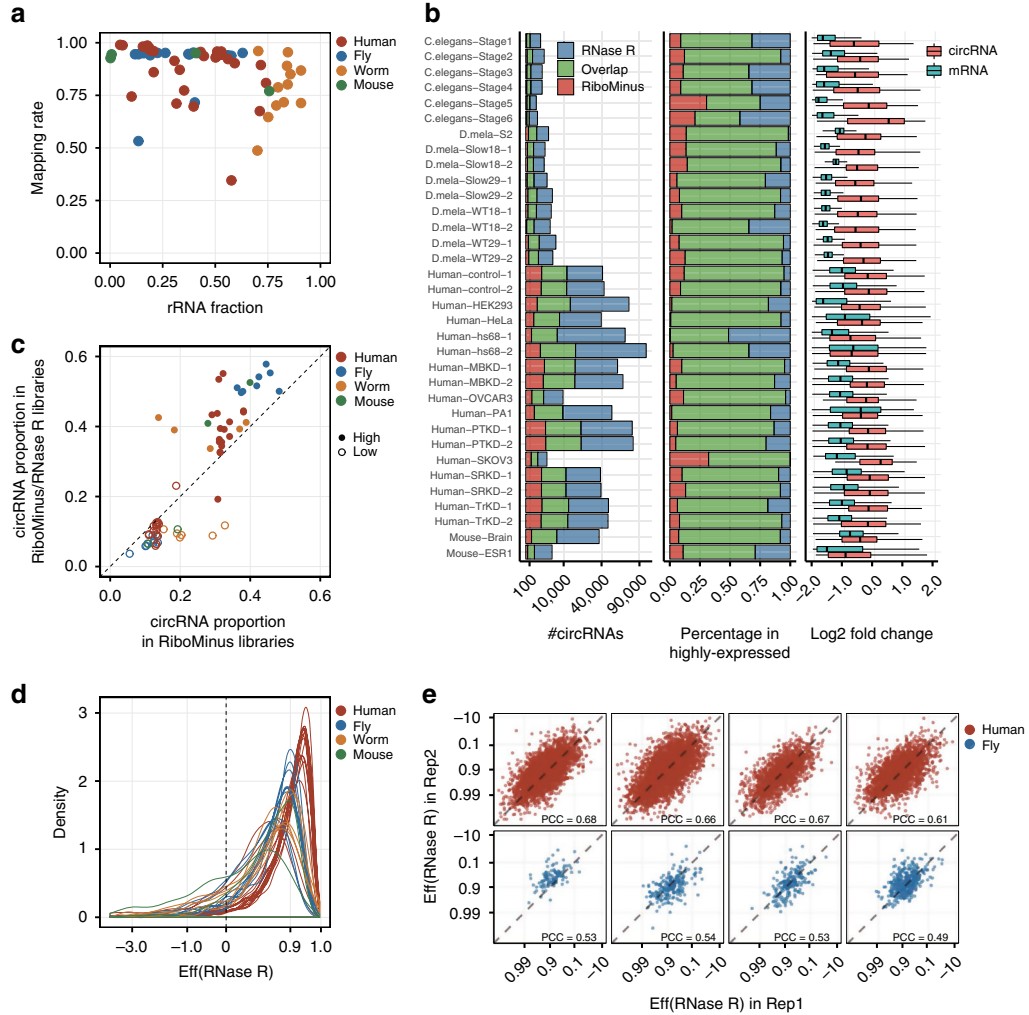

**Fig. 1 Challenges in quantifying the expression of circular RNAs. a** The fraction of ribosomal RNA reads in 63 RNA-seq datasets from four different species. x- and y-axis represent the fraction of reads mapped to rRNA sequences and the reference genome in all sequencing reads, respectively. **b** RNase R treatment affects circRNA numbers and their expression levels. Left: total number of detected circRNAs in RiboMinus and RiboMinus/RNase R-treated samples. Middle: percentage of highly expressed circRNAs detected in both RiboMinus and RiboMinus/RNase R samples. Right: Boxplot shows the change of circRNA and mRNA expression levels after RNase R treatment. Expression levels of circRNAs were measured as the number of BSJ reads of each circRNA divided by the total number of BSJ reads. Gene expression levels were measured by TPM (transcript per million). **c** Proportion of highly (top 50%) and lowly (bottom 50%) expressed circRNAs reads in RiboMinus and RiboMinus/RNase R data. RNase R treatment tends to enrich highly expressed circRNAs but reduce lowly expressed circRNAs. **d** Density distribution of RNase R enrichment coefficient for circRNAs in four species. The vertical dashed line at $x = 0$ indicates a null enrichment effect of circRNAs. **e** Replication of RNase R enrichment coefficient in two experimental replicates of eight RNA-seq datasets. X- and y-axis represent the log transformed $(1-x)$ of RNase R treatment coefficient.

mean enrichment coefficient of 0.95. In contrast, the peak of RNase R coefficient in RNA-seq samples of other species is generally lower than 0.9. To further investigate the random effect in RNase R treatment, we compared the enrichment coefficient of circRNAs in experimental replicates. The enrichment effect exhibited a high correlation in different experimental replicates in general, but when focusing on a certain circRNA, it showed a lower consistency (Fig. 1e). Taken together, these findings show that the quantification of circRNAs is affected by their abundance, rRNA removal rate, and RNase R treatment efficiency, which requires more efficient algorithms to tackle these problems.

**CIRIquant—accurate quantification of circRNAs.** Current computational approaches on circRNA identification are mainly based on the detection of BSJ reads. Most of these methods (e.g.,

CIRCexplorer2, DCC, find_circ) rely on specific RNA-seq aligners to detect anchor sequences or map fusion reads, and then scan mapping results for circular transcript identification. However, these strategies exhibited significant shortcomings in quantifying circRNA expression from RNA-seq data, as the RNA-seq aligners they used are not designed for mapping reads with a BSJ signal, especially for those spanning multiple junction sites[8,25]. Other tools either use BWA-MEM to obtain split mapping position of junction reads[11,12], or employ a model-based statistical algorithm to quantify circRNA expression levels[10], of which the sensitivity and accuracy for circRNA detection and quantification are dependent on the threshold and model selection. Here, we proposed a new and efficient approach (Fig. 2a) to accurately identify and quantify both linear and circular transcripts across BSJ from transcriptomic data. First, RNA-seq reads were aligned to the reference genome using HISAT2, and CIRI2[12] or other circRNA detection tools were applied to identify putative circRNAs. To

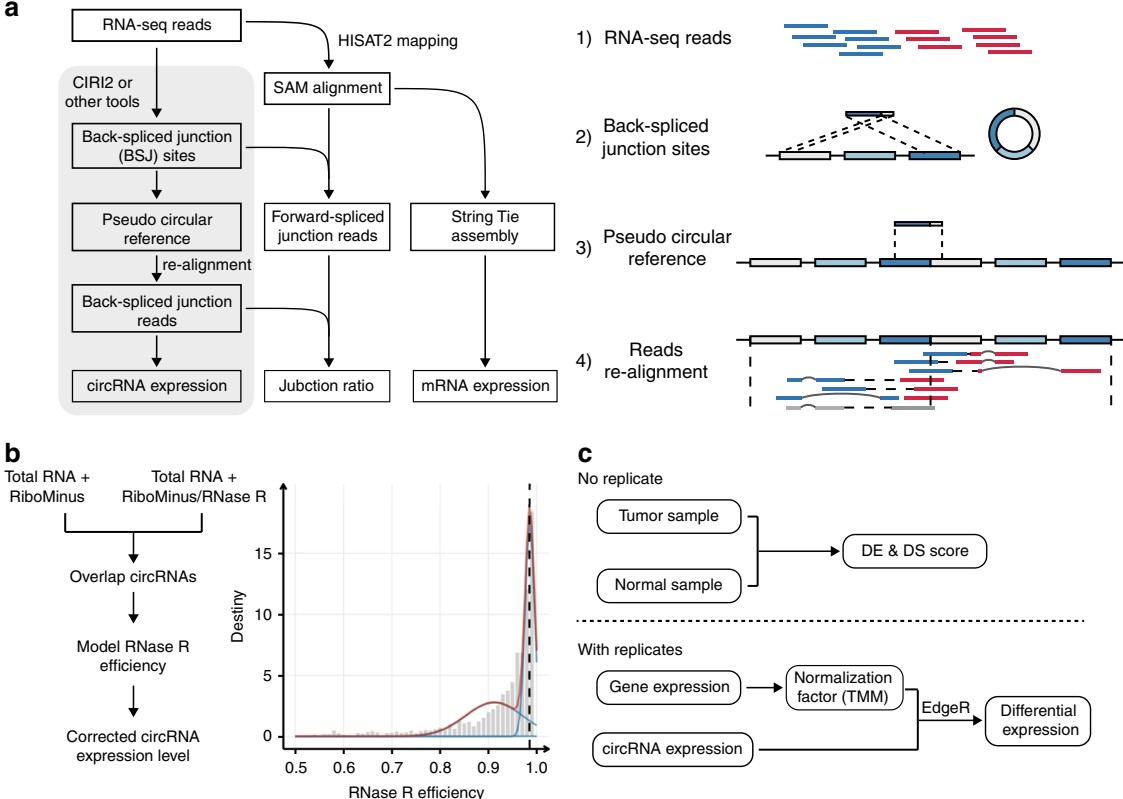

**Fig. 2 Overview of the CIRIquant pipeline. a** Workflow of circRNA quantification. CircRNAs are first identified by CIRI2 or any other computational methods. Then, CIRIquant detects back-spliced junction reads using re-alignment of RNA-seq reads against the pseudo-circular reference. Expression level and junction ratio of circRNAs are calculated using BSJ reads and FSJ reads, and stringTie is used to estimate gene abundance. **b** The correction of RNase R treatment efficiency. Overlapped circRNAs detected in both RiboMinus and RiboMinus/RNase R data are used for enrichment efficiency calculation and a gaussian mixture model (GMM) is used for model fitting. The fitted model is used as prior distribution for the correction of circRNA expression values. **c** Differential expression analysis of circRNAs. Top: For paired samples without replicates, DE-score and DS-score is directly calculated using generalized fold change. Bottom: For paired samples with biological replicates, gene expression values are used to calculate normalization factor, then the modified edgeR pipeline is used for statistical significance test.

accurately quantify the expression level of circRNAs and filter false-positive BSJs, we generated a pseudo circRNA reference sequence by concatenating two full-length sequence of the BSJ region. Then, candidate circular reads were re-aligned against this pseudo reference, and BSJ reads were determined if they could be linearly and completely aligned to the BSJ region. Furthermore, by combining the alignment results against the reference genome and the pseudo-reference sequences, we could determine the junction ratio for each circRNA by calculating the percentage of circular splice junction reads across the BSJ. Finally, canonical RNA-seq data analysis pipeline[23,26] was used to obtain transcript-level expression information[27].

For RNase R-treated RNA-seq data, the circRNA BSJ expression values cannot be directly used for comparative analysis due to the uneven efficiency of RNase R treatment in different studies (Fig. 1d). Hence, we implemented a Gaussian mixture model to fit its efficiency distribution (Fig. 2b), and then used the fitted model as posterior distribution for RNase R coefficient correction. For differential expression analysis of circRNA, we proposed two strategies to evaluate both differential expression (DE) and differential splicing (DS) of circRNAs in case and control samples. When no biological replicate is available, we calculated DE and DS score for circRNAs using generalized fold change, utilizing both fold change and variance information, which provide more meaningful rankings. With biological replicate samples, statistical test was performed to evaluate the

significance of change in circRNA expression values and junction ratios. To infer the true difference in circRNA expression between samples, we implemented trimmed mean of logarithm fold changes (TMM) normalization using gene expression data to remove systematic batch effects[28] (Fig. 2c). Consequently, generalized linear models in edgeR[29] is applied to determine whether a circRNA is significantly differentially expressed across experimental conditions and exact rate-ratio test is used as significance test for difference in circRNA junction ratio.

**Simulation studies**. To evaluate the performance of existing algorithms on circRNA quantification, we used CIRI-simulator[12] to generate simulated datasets for performance comparison. We firstly simulated RNA-seq reads with different read lengths ranging from 100 bp to 250 bp, and then applied CIRI2, CIRCexplorer2[16], DCC[13], find_circ[2] and KNIFE[30] to assess their sensitivities on circRNA detection. For most tools except CIRI2, the detection sensitivity was reduced with the increase of sequencing read length. Therefore, to get better performance for all tools, we chose simulated data with paired-end 100 bp reads and circRNAs under empirical expression distribution. We applied each of the five methods to detect circRNAs from the simulated data, and then used the predicted circRNA coordinates as input for CIRIquant to filter false positives and quantify circRNA expression. Next, we calculated the Pearson correlation

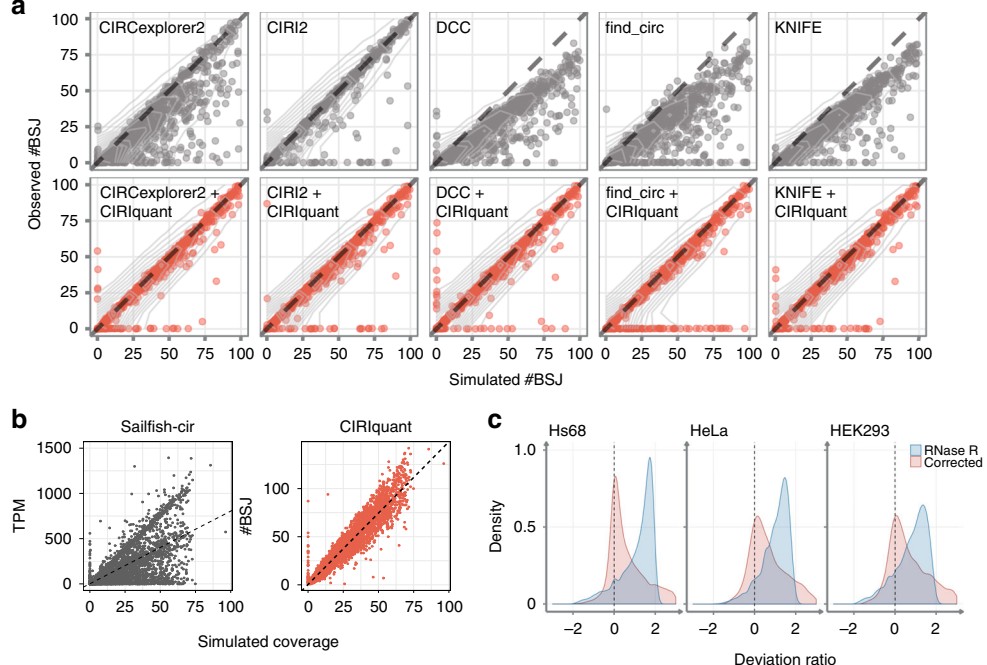

**Fig. 3 Performance of CIRIquant on simulated data. a** The correlation between simulated and estimated number of BSJ reads. CIRIquant was performed with predicted circRNA junction sites using five different algorithms. The correlation before and after CIRIquant correction are shown separately. Obviously, for all these tools, the inclusion of CIRIquant can significantly improve circRNA quantification and remove false-positive predictions. **b** Comparison of Sailfish-cir and CIRIquant on the simulated dataset. TPM of identified circRNAs was used to evaluate the performance of Sailfish-cir. *x*-axis represents the simulated coverage of each circRNA, and the fitted linear-regression model was plotted as the dashed line. **c** Density distribution of deviation ratio of circRNA expression levels between RiboMinus and RiboMinus/RNase R samples. The distribution of deviation ratio derived from uncorrected BSJ reads and expression values after CIRIquant correction are shown in blue and red, respectively. After the correction of RNase R treatment efficiency by CIRIquant, there is no much difference on circRNA quantification between RiboMinus and RiboMinus/RNase R samples.

coefficient between the number of predicted BSJ reads and the number of simulated BSJ reads for identified circRNAs. As shown in Fig. 3a, among these methods, CIRI2 achieved the best performance (PCC = 0.97), while the correlation coefficient of the other tools ranged from 0.87 (CIRCexplorer2) to 0.92 (KNIFE). After the adjustment of BSJ reads by CIRIquant, the correlation coefficient for all the five methods improved remarkably (Supplementary Fig. 2). It should be noted that besides more accurate quantification, CIRIquant exhibited the lowest false discovery rate (FDR) for circRNA identification, which should be attributed to its re-alignment of putative BSJ reads against the pseudo-reference sequence. Sailfish-cir[10] implemented a similar strategy to quantify circRNA expression by transforming circular transcripts into pseudo-linear transcript and then employed Sailfish[31] to estimate the expression values of both circular and linear transcripts. Considering that Sailfish-cir only estimated circRNA expression levels measured as transcripts per million (TPM) without counting BSJ reads count, we calculated the Pearson correlation coefficient between the simulated coverage and the predicted TPM by Sailfish-cir or BSJ reads number by CIRIquant. As shown in Fig. 3b, CIRIquant (PCC = 0.947) achieved much better performance than Sailfish-cir (PCC = 0.618) on the simulated dataset. A major reason is that Sailfish-cir relied on the unique sequence in circular transcripts to distinguish them from linear transcripts, which is only applicable to circRNAs with low overlap with their parental mRNA transcripts. Additionally, we obtained the qRT-PCR data from a previous study[30] to assess the consistency of predicted and experimental results. As shown in Supplementary Fig. 3, CIRIquant clearly outperformed all other tools (correlation coefficient $r = -0.79$, $p = 3.8 \times 10^{-18}$). Taken together, these results demonstrated that CIRIquant can achieve

better performance in circRNA quantification on both simulation and experimentally validated datasets.

**RNase R treatment correction**. The inconsistency of RNase R digestion efficiency in circRNA library preparation for different samples may lead to biased circRNA expression quantification. To evaluate the performance of RNase R efficiency correction in CIRIquant, we applied this method in three RNA-seq datasets of human cell lines generated in previous studies[17,18,22] and each dataset consists of both RiboMinus and RiboMinus/RNase R libraries. We implemented Gaussian mixture model to fit the distribution of RNase R digestion coefficient, and then used its posterior distribution to estimate the original read counts for each circRNA before RNase R treatment. Subsequently, we calculated the deviation ratio between estimated CPM before and after RNase R correction. As shown in Fig. 3c, the distribution of deviation ratio in corrected CPM exhibited much less dispersion level than those without RNase R efficiency correction. In RiboMinus/RNase R data, the expression levels of circRNAs tended to be overestimated due to the enrichment of circRNAs by RNase R, and the correction of RNase R treatment efficiency could greatly minimize the bias between RNase R-treated and untreated samples. To experimentally validate the reliability of RNase R correction, 20 quantitative real-time RT-PCR experiments were performed on five randomly selected circRNAs in four RiboMinus libraries (Supplementary Fig. 4). The root-mean-squared error (RMSE) was used to measure the deviation of qRT-PCR results and the expression level of circRNAs predicted in the RiboMinus/RNase R libraries. After RNase R correction, CIRIquant significantly reduced the bias

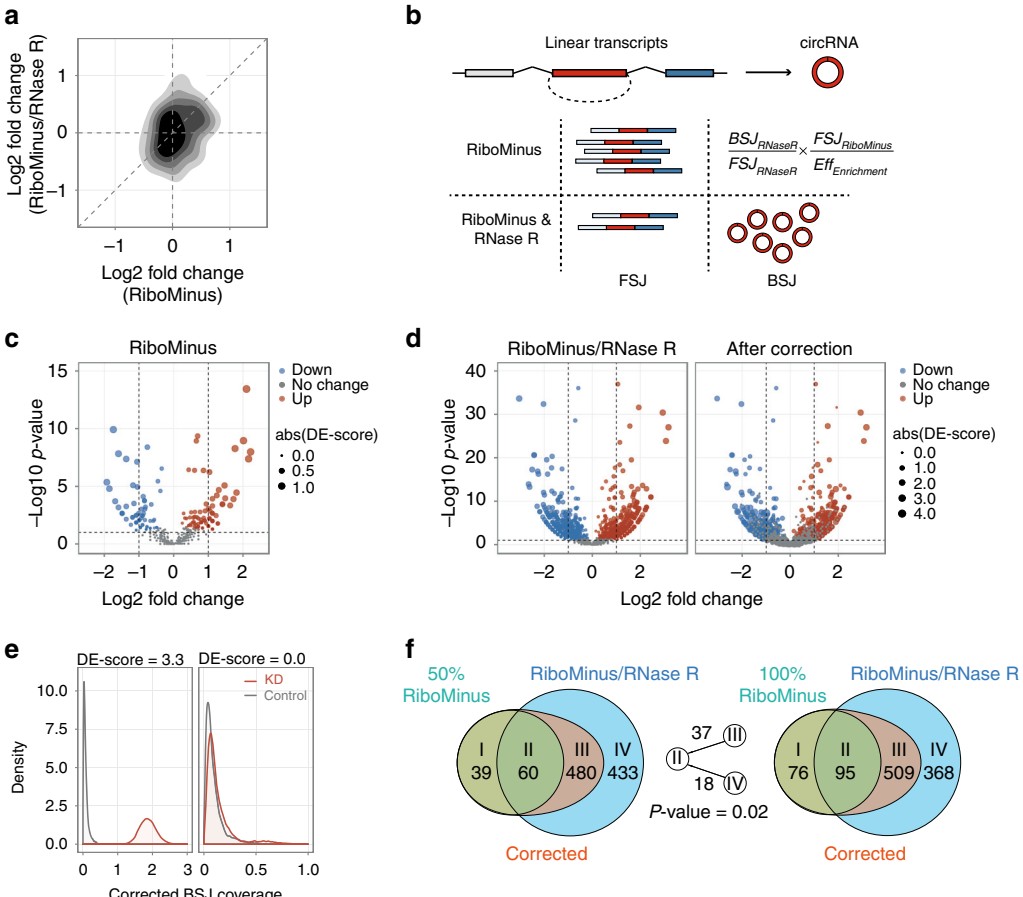

**Fig. 4 CircRNA differential expression analysis after RNase R correction. a** The correlation of log2 fold change after TRA2B knockdown in RiboMinus and RiboMinus/RNase R-treated libraries. **b** The schematic workflow of RNase R treatment efficiency correction. Firstly, the BSJ and FSJ reads counts from RiboMinus/RNase R sample were obtained. With the posterior distribution of RNase R enrichment coefficient, the corrected BSJ reads count in RiboMinus sample can be calculated using the equation shown above. **c** Volcano plot of −log10 p-value (y-axis) against log2 fold change (x-axis) of circRNAs in RiboMinus samples. Size of points indicates DE-score values calculated by CIRIquant. Significantly upregulated and downregulated circRNAs are labeled in red and blue, respectively. **d** The volcano plot of circRNAs in RiboMinus/RNase R libraries before and after CIRIquant correction. Red and blue nodes represent up and downregulated circRNAs, respectively, with their node sizes representing DE scores. **e** CircRNA with relatively higher change in prior probability distribution of the number of BSJ reads will have a more significant DE score. Red and gray lines represent posterior distribution of corrected circRNA expression values before and after TRA2B knockdown. **f** Overlap of significantly differentially expressed circRNAs after and before RNase R correction. The whole and randomly sub-sampled of 50% RiboMinus RNA data are used as control set for RNase R treatment correction in two Venn diagrams. DE-circRNAs are divided into four categories: (I) RiboMinus-specific DE-circRNAs, (II) DE-circRNAs in both RiboMinus and RiboMinus/RNase R-corrected groups, (III) RiboMinus/RNase R-corrected DE-circRNAs, (IV) Dropped DE-circRNAs after RNase R correction. Numbers between the two Venn diagrams show the amount of circRNAs in type III and type IV that can be re-classified into type II when increasing the data size from 50 to 100%. The type III DE-circRNAs are more likely to be re-classified into type II than those in type IV (p-value = 0.02, Chi-squared test), demonstrating the reliability of RNase R correction by CIRIquant.

induced by RNase R treatment (RMSE = 0.006) and achieved much better performance than all the other tools (Supplementary Fig. 4).

To further investigate the effect of RNase R treatment on differential expression analysis, we performed RNA interference-mediated knockdown of TRA2B in HeLa cells, where TRA2B, an important sequence-specific serine/arginine splicing factor, was depleted using shRNA designed to minimize off-target effects. We isolated total RNA before and after TRA2B knockdown, and then divided each sample into two fractions, where ribosomal RNA depletion and RNase R digestion were performed in one fraction (termed as RiboMinus/RNase R) and the other fraction was only treated by ribosomal RNA depletion (termed as RiboMinus). 52–88 million paired-end 100 bp reads were generated for these RNA-seq libraries. Consequently, we employed CIRI2 and CIRIquant to identify circRNAs and quantify their expression

levels, and then calculated the logarithmic fold change of circRNA expression levels after knockdown of TRA2B for RiboMinus and RiboMinus/RNase R libraries, respectively. We observed that for circRNAs which can be detected in both datasets, the log2-fold change between RiboMinus and RiboMinus/RNase R groups showed a low correlation (Fig. 4a, Pearson correlation coefficient = 0.21, p-value = 0.005), indicating that RNase R treatment should cause significant bias in subsequent analysis. For instance, we observed an inconsistent pattern of expression level change for several circRNAs after TRA2B knockdown between RiboMinus and RiboMinus/RNase R datasets (Supplementary Fig. 5). In the RiboMinus dataset, the expression of circPRKD3, circZBTB46, circVAPA, and circCHSY1 increased significantly after TRA2B knockdown, while circATXN7 showed a decrease in expression. For the RiboMinus/RNase R dataset, however, an opposite trend was

observed for these five circRNAs. Such contradictory results reflect the non-negligible randomness effect of RNase R enrichment in circRNA studies. Therefore, correction for RNase R treatment efficiency is essential for circRNA differential expression analysis.

To systematically evaluate the potential bias caused by RNase R treatment in differential expression analysis, we employed two different strategies to process RiboMinus and RiboMinus/RNase R datasets. Firstly, for the RiboMinus data without RNase R treatment, CIRIquant calculated the differential expression (DE) score using the generalized fold-change method adjusted from GFOLD[32], in which differentially expressed circRNAs were ranked by taking into account both fold change and variance of the posterior distribution of log2 fold change (Fig. 4c). As shown in the volcano plot, DE score by CIRIquant showed a strong consistency between log2 fold change and P-value in the RiboMinus data. A DE score of zero means no significance in expression change, and circRNAs with higher DE score indicate a relative greater change of expression level and smaller p value compared to those with smaller DE score. Secondly, for the RiboMinus/RNase R data, the distribution of RNase R digestion efficiency was estimated using circRNAs found in both RiboMinus and RiboMinus/RNase R data. Then, a Gaussian mixture model (GMM) was used to fit the distribution. With the fitted model as prior distribution, the original BSJ reads count in the RiboMinus data can be inferred from the number of BSJ reads in the RiboMinus/RNase R data and the forward-spliced junction (FSJ) reads in both datasets (Fig. 4b). Finally, the posterior distribution of read counts after correction in case and control samples were used for DE-score calculation (Fig. 4e). Compared to DE score simply derived from the RiboMinus/RNase R data, additional step of RNase R correction can filter out a majority of circRNAs with relatively low fold change and P-value, which are considered to be significantly differentially expressed before RNase R correction (Fig. 4d). Moreover, the correction for RNase R treatment can also affect the ranking of circRNAs with non-zero DE score.

To further validate the reliability of this correction method, we randomly sampled half of reads from the RiboMinus data, and used this subset of sequences to identify circRNAs and then compared them with the same RNase R-treated sample to calculate DE scores. We divided the identified DE circRNAs into four categories: (i) RiboMinus-specific DE-circRNAs (circRNAs that were found to be differentially expressed only in the RiboMinus dataset), (ii) DE-circRNAs in both RiboMinus and RNase R-corrected groups, (iii) DE-circRNAs solely identified in the RNase R-corrected group, (iv) circRNAs that were filtered out by RNase R correction. With the increase of reads in the control set from half of the data to all reads, we observed 37 circRNAs in type III and 18 circRNAs in type IV that were re-classified into type II (Fig. 4f), which indicated that after RNase R correction, the DE-circRNAs are more reliable compared to those filtered out by our method ($p = 0.02$, Chi-squared test). Collectively, these results demonstrate that the RNase R correction implemented in CIRIquant can efficiently filter false positives and generate more reliable differential expression analysis.

**Identification of linear-circular isoform switching events.** To determine the ratio of circRNA to its parental linear transcript, CIRIquant takes advantage of the two-step read alignment strategy, including canonical RNA-seq read alignment and the re-alignment to pseudo-reference sequence. Therefore, the ratio of BSJ reads to canonical junction reads for a given circRNA at the junction site can be accurately determined. We applied CIRI2, DCC and Sailfish-cir on the same simulated datasets described

above, and found that these methods varied greatly in junction ratio estimation (Fig. 5a). Sailfish-cir and DCC tended to underestimate junction ratio, largely due to their relatively low sensitivity in detecting BSJ reads, which is consistent with the simulation results (Fig. 3a). CIRI2, however, slightly over-estimated the junction ratio for a majority of circRNAs in this simulated data. A possible explanation is that CIRI2 used BWA-MEM for split mapping of BSJ reads, in which the aligner is not designed for gapped alignment of RNA-seq reads and may lead to an underestimation of forward-spliced junction reads. As shown in Fig. 5a, CIRIquant, which utilized the pseudo-reference alignment strategy, achieved notably high performance in junction ratio estimation (PCC = 0.97) and outperformed all the other approaches.

To further demonstrate the applicability of CIRIquant on differential expression analysis, we knockdown three well-known splicing factors (MBNL1, PTBP1, and TRA2B) in HeLa cells using specifically designed shRNAs, and then isolated total RNA of three knockdown samples and a control sample with mock treatment for transcriptome sequencing. We applied CIRI2 and CIRIquant to detect and quantify circRNAs, and then performed differential expression analyses using both expression levels and junction ratios (Fig. 5b). DE score of circRNAs were calculated as previously described, to assess differential expression levels. However, for the change of junction ratio, we used a similar method to calculate the differential splicing (DS) score (see the Methods section). To experimentally validate the accuracy of circRNA junction ratio estimation by CIRIquant, 60 quantitative real-time RT-PCR assays were performed on five randomly selected circRNAs and their parental linear transcripts in the control set and three knockdown libraries. For each circRNA, outward primers were used to amplify BSJ region, while two pairs of inward primers targeting 5' and 3' cirexons were designed to determine junction ratio of these circRNAs. The junction ratio of circRNAs obtained from qRT-PCR and CIRIquant showed a high level of consistency (Supplementary Fig. 6), indicating the reliability of CIRIquant on the determination of circRNA junction ratios.

Based on differential expression analysis, we found that 162, 346 and 97 DE-circRNAs in the three knockdown samples, respectively. Then we calculated the DS scores and found that DS-circRNAs and DE-circRNAs showed a high concordance (Fig. 5c). Nevertheless, we observed a certain fraction of circRNAs that were only identified by using DS or DE score, which we termed as DS- and DE- specific circRNAs, respectively. The fraction of DE-specific and DS-specific circRNAs varied in the three knockdown datasets. While PTBP1 and TRA2B knockdown resulted in more DS-specific circRNAs, knockdown of MBNL1 only gave rise to 36 DS-specific compared to 60 DE-specific circRNAs (Fig. 5d), which indicates that the knockdown of these splicing factors may affect circRNA biogenesis through different mechanisms. To investigate the influence of splicing factor knockdown on circRNAs and alternative splicing of mRNA transcripts, we chose the PTBP1 knockdown sample, which has the largest number of DE and DS circRNAs, for further analysis. Gene ontology enrichment analysis was performed on these DE- and DS-circRNAs and differentially expressed genes by GFOLD[32] (Fig. 5e). The nitric oxide and interleukin-1 related processes were enriched in differentially expressed genes, in contrast to the DNA repair and phosphorylation process enriched in DE&DS circRNAs (Supplementary Table 2).

In different cell types or in response to stimuli, genes can drastically change the relative abundance of different isoforms, often referred to as isoform switching, which may have substantial biological impact[33]. Here, we defined linear-circular isoform switching (LC-switching) as that occurring between

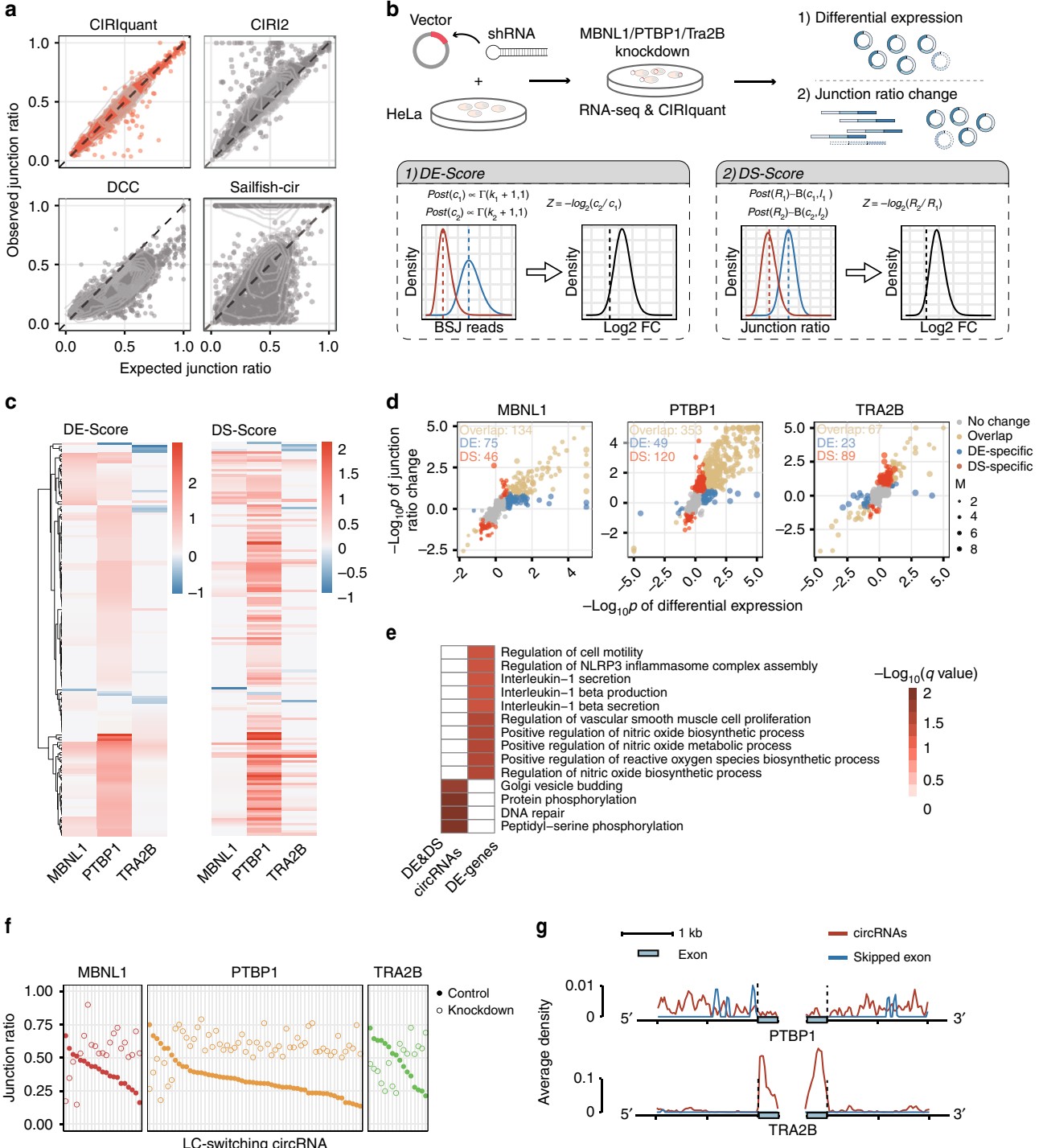

**Fig. 5 Differentially spliced circRNAs after knockdown of three splicing factors. a** The correlation between simulated and estimated junction ratio by CIRIquant and three other methods. *x*- and *y*-axis represent the expected and estimated junction ratio of circRNAs, respectively. For DCC, the junction ratio was calculated using the results from LinearCount and CircRNACount output. For Sailfish-cir, the TPM of circRNAs and corresponding host genes were used for junction ratio calculation. **b** Schematic view of experimental design for shRNA induced knockdown of three splicing factors (MBNL1, PTBP1, and Tra2B) and the calculation of DE- and DS- score. Specially designed shRNAs were used to knockdown target genes. Then, RNA-seq were performed for control and knockdown groups, and circRNA expression levels and junction ratio were estimated by CIRIquant for further differential expression analysis. **c** Heatmap of DE score and DS score in three knockdown datasets. Only circRNAs significantly changed in at least one sample are shown in the heatmap. **d** Overlap of significantly changed circRNAs. *x*- and *y*-axis represent −log10(*p* value) from DE-analysis and rate-ratio test using junction ratio of circRNAs in two different conditions, respectively. Downregulated *p*-values are assigned as negative values. DE-specific and DS-specific circRNAs are highlighted in blue and red, respectively. **e** Gene ontology enrichment analysis of differentially expression genes and all significantly changed DE- and DS-circRNAs. The logarithm of FDR was calculated in Enrichr, then plotted in heatmap for comparison. **f** Junction ratio changes of LC-switching circRNAs after MBNL1/PTBP1/TRA2B knockdown. **g** Average depth of binding sites in flanking regions of differentially spliced ES exons and significantly changed circRNAs using CLIP-seq results from iCLIPdb. A sliding window analysis with 8 bp windows was performed for density calculation. Vertical axis represents the average binding site per shifted window.

circular transcript and its parental linear transcript. A stringent threshold was set to detect highly confident LC-switching events, where the junction ratio should show a dramatic change in different samples (Δ Junction ratio > 0.3) and the LC-switching circRNAs should change from the major splicing isoform (junction ratio > 0.5) to the minor isoform (junction ratio < 0.5), or vice versa. As shown in Fig. 5f, we found multiple linear-circular isoform switching events in these three knockdown samples (58 in PTBP1, 21 in MBNL1 and 16 in TRA2B, Supplementary Table 3 and Supplementary Fig. 7), suggesting a competitive regulation of host gene canonical splicing and circRNA production after the knockdown of these splicing factors[34]. We downloaded public iCLIP datasets of PTBP1 and TRA2B from CLIPdb[35,36], and then detected their binding sites from the genomic regions adjacent to (i) DE and DS circRNAs (ii) alternatively skipped exons detected by MISO[37]. The average coverage of binding sites was calculated for each base on the flanking regions of BSJ and exon boundaries (Fig. 5g). In the PTBP1 and TRA2B-knockdown data, we observed multiple distinct binding peaks within back-spliced circular exons and their flanking 3 kb regions. For PTBP1, skipped mRNA exons showed a high peak of binding site density in 5' exon boundary, whereas DE- and DS-circRNAs have continuous binding sites in both upstream and downstream intronic regions. In contrast, for TRA2B, the DE- and DS-circRNAs enriched TRA2B binding sites in exonic region, while no significant peak were observed for skipped mRNA exons. For skipped mRNA exons, the binding site preference of PTBP1 and TRA2B is consistent to previous studies[38,39]. Such a significant difference in binding specificity for circRNAs may explain how splicing factor PTBP1 and TRA2B bind with pre-mRNA and regulate circRNA splicing. Altogether, we proposed a novel approach to investigate linear-circular isoform switching events in circRNA studies, which help us understand the mechanism of circRNA biogenesis and unveil the regulation of competitive splicing between circRNA and their linear counterparts.

**Extensive transcriptomic changes in hepatocellular carcinoma**. To systematically investigate the expression profile of circRNAs in hepatocellular carcinoma (HCC), we applied CIRI2 and CIRIquant to 40 RNA-seq datasets of normal and tumor liver samples derived from HCC patients[40] and a total number of 46,984 circRNAs were detected and quantified in these samples. We firstly performed dimensionality reduction using mRNA expression level, circRNA expression level and junction ratio of circRNAs. The tumor and adjacent normal samples were clearly distinguished using all three features (Fig. 6a), indicating that like mRNA, circRNA and junction ratio can also be served as potential biomarkers to distinguish tumor from normal tissue samples.

We further performed differential expression analyses using all three features mentioned above, and then classified genes hosting circRNAs into three categories: differentially expressed genes, host genes of DE-circRNAs and host genes of DS-circRNAs. In total, 1159 genes were significantly changed between normal and tumor samples, while there were only 459 genes hosting DE-circRNAs and 284 genes hosting DS-circRNAs (Fig. 6b). It should be noted that there were very few overlapped genes among these three categories, indicating that DE- and DS-circRNAs may play different roles in the regulation of tumor development (Supplementary Fig. 8). Then, we plotted the top 25 DE-circRNAs, of which only 3 were also detected in top 25 DS-circRNAs (Fig. 6c). Notably, several well-characterized cancer-related genes (e.g., *ZKSCAN1*, *ABCB4* and *CYP2C8*) were found to generate circRNAs with significant changes in tumor samples. Among these genes, we found circZKSCAN1 and circSMARCA5 are both downregulated in HCC tumor samples, which confirmed previous findings[41,42].

To further explore the change in circRNA expression patterns from different aspects, we calculated the junction ratio for each circRNA and circular transcript usage (CTU) for circRNA host genes. For each circRNA host gene, only two major circular transcripts were considered, and the proportion of the circRNA with the longest junction site distance was used to determine switching in circular transcript usage. In order to get a convincing result, only circRNAs that expressed in more than half of samples in both tumor and normal groups were used for detecting CTU-switching events. As shown in Fig. 6d, most LC-switching circRNAs showed significant decrease of junction ratio in tumor compared to normal samples, while CTU-switching circRNAs exhibited both up and downregulated patterns (Supplementary Table 4). Moreover, we also observed a number of LC-switching and CTU-switching events in two additional datasets (Alzheimer's Disease and Renal cell carcinoma), which strongly supported our findings (Supplementary Fig. 9). Next, we investigated the tissue specific expression pattern of all LC/CTU-switching circRNAs, measured by the relative expression level of circRNA in each tissue from circAtlas[43] (Fig. 6e). Several circRNAs were observed with highly conserved expression in all tissues, suggesting that these circRNAs may play conservative roles in certain biological process. Moreover, we noticed that hsa-intergenic_006404, an intergenic circRNA from chromosome 5, was also recognized as the LC-switching circRNA in the HCC dataset. It exhibited a significant decrease in junction ratio in tumor samples, as well as a liver-specific expression pattern (Fig. 6f). With quantitative real-time PCR (qRT-PCR) and Sanger sequencing, we confirmed the expression level of hsa_intergenic_006404 in 14 pairs of tumor and adjacent normal samples, and also validated its BSJ sequence. As most circRNAs are derived from genic region, we further studied the epigenetic feature at the genomic region of hsa_intergenic_006404, and found that this circRNA was derived from an enhancer region, and its flanking region exhibited an enrichment of H3K27 acetylation in all 7 ENCODE cell lines[44,45] (Fig. 6f), which is thought to represent the active regulation of transcription[46]. Additionally, we observed a large overlap with transcription factor CHIP-seq peaks and DNase I hypersensitivity clusters in its BSJ region (Supplementary Fig. 10). Altogether, as a liver-specific circRNA, hsa_intergenic_006404 may play an important role in the development of HCC.

Among the CTU-switching circRNAs in 40 HCC RNA-Seq samples, circMTUS1 exhibited a significant increase of CTU in tumor samples compared to normal ones, while circVWA8 switched from major to minor isoform in tumor samples (Fig. 6g). Considering that RNA binding proteins (RBPs) are involved in the biogenesis of circRNAs[1,34], we speculate that RBPs may play important roles in those CTU-switching events by affecting the formation of different circRNA transcripts. Thus, we calculated the correlation of RBP expression levels and major isoform CTU of differential circular usage (DCTU) genes. A total 21 RBPs were selected from previous study[34], and the logarithm of $p$-values from Spearman correlation test was plotted (Fig. 6h). As expected, PTBP1 was correlated with more DCTU genes compared to TRA2B and MBNL1, which is consistent to our previous results. Taken together, these findings demonstrate that by using both circRNA junction ratio and circular transcript usage, CIRIquant can provide novel angles in circRNA analysis, enhancing our ability to explore extensive transcriptomic changes in tumorigenesis.

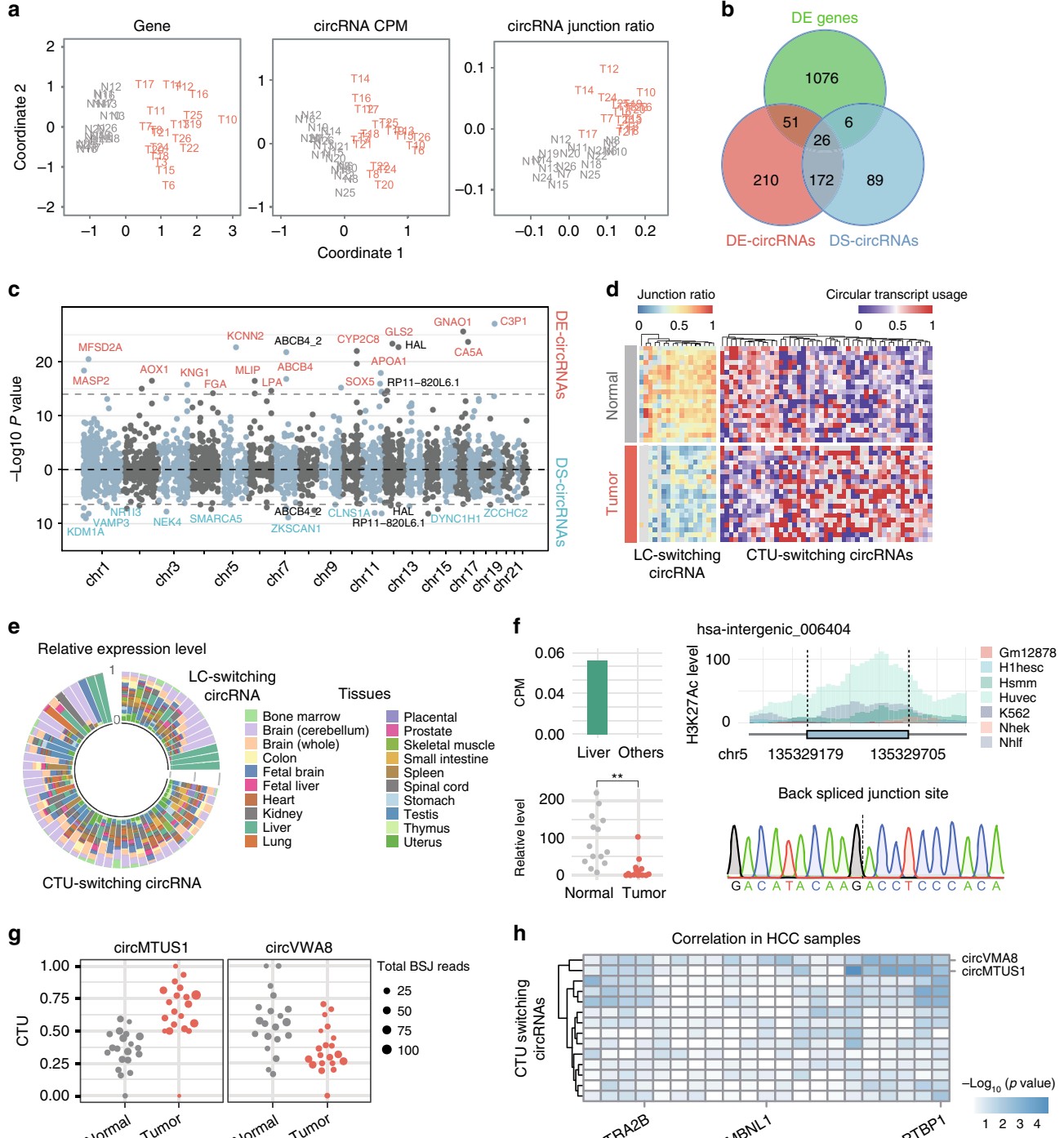

**Fig. 6 Differentially expressed circRNAs in hepatocellular carcinoma.** **a** Multi-dimensional analysis using gene expression level, circRNA expression values and junction ratios. Distances between samples are calculated based on log2 fold change. Tumor and normal samples are shown in red and gray, respectively. **b** Overlap of significantly changed genes and host genes of DE-circRNAs and DS-circRNAs. **c** Manhattan plot of p-values from DE- and DS-circRNAs. Horizontal dashed lines indicate top 25 most significant DE- and DS-circRNAs. DE-specific and DS-specific circRNAs are shown in red and blue, respectively. CircRNAs that are shared by DE and DS groups are shown in black. **d** Hierarchical clustering of linear-circular (LC) switching and circular transcript usage (CTU) switching circRNAs. The junction ratio and CTU of these circRNA in 20 pairs of HCC samples were plotted, respectively. **e** Tissue specific expression level of LC-switching and CTU-switching circRNAs. The relative expression level represents the proportion of expression value in specific tissue compared to all 20 tissues. **f** An instance of LC-switching circRNAs, hsa-intergenic_006404. The bar plot indicates the liver-specific expression pattern of this circRNA. The beeswarm plot shows its expression level in 14 pairs of tumor and normal tissues measured by qPCR. The H3K27Ac marks of upstream and downstream 300 nt region are plotted, indicating the potential biogenesis mechanism of hsa-intergenic_006404 from intergenic region. **g** Top two CTU-switching circRNAs that are correlated with the most RBPs. **h** The correlation between RBP expression level and CTU of CTU-switching circRNAs. The color gradient in the heatmap represents the −log10 p value from spearman correlation test.

## Discussion

Accurate quantification of circRNAs is a challenging aspect in circRNA related studies. Although several computational methods have been developed for circRNA detection, the reliability of the output is a crucial factor for further analysis. In recent studies, the sensitivity and specificity of currently available circRNA detection algorithms have been thoroughly assessed[9,47]. However, the precision for circRNA quantification is still not well characterized. In order to accurately quantify the expression of circRNAs, we developed CIRIquant, a versatile toolkit for circRNA quantification analyses. By utilizing RNA-seq aligner to convert the split mapping of junction reads to the spliced alignment of linear reads, CIRIquant exhibited a significantly increased sensitivity on back-spliced junction reads detection. In CIRIquant, a generalized fold change of BSJ reads count and junction ratio was implemented for differential expression analysis, which proved to be an effective ranking method for differentially expressed circRNAs both in samples with/without replicates. Through comprehensive evaluations on both simulated and real datasets, along with qRT-PCR validation, we demonstrated that CIRIquant exhibits a high efficiency and reduced false-positive rate on circRNA detection and quantification compared with previous methods.

In current circRNA studies, there are many factors that affect the reliability of circRNA characterization and quantification. Although a few algorithms have been developed for direct estimation of circRNA expression levels[2,10,12,13,16], most computational tools for circRNA characterization simply use the number of BSJ reads to quantify circRNA expression. Considering the variation of read length, library size and mapping rate in different datasets, normalization of circRNA expression values is necessary for next-step analysis. For circRNA sequencing without RNase R treatment, due to the fraction of circRNA is usually lower than 1% in transcribed non-ribosomal RNAs (Fig. 1), it requires a large amount of reads to reach optimal detection of circRNAs. Considering that the number of detected circRNAs in different samples varies from a few hundred to tens of thousands, the sum of BSJ reads is obviously not a proper denominator for normalization in different studies. Instead, counts per million mapped reads (CPM) can be used to achieve an unbiased estimation of circRNA expression. In addition, for RNase R-treated samples, a randomized enrichment effect was observed, as well as the increased fraction of highly expressed circRNAs, which suggest that RiboMinus library without RNase R treatment is more suitable for expression level analysis of circRNAs.

In addition, the efficiency of RNase R enrichment for circRNAs is usually distinct among different samples. The enrichment coefficient of the same circRNA in different experimental replicates is also lower than expected, suggesting that the correction of RNase R depletion is essential when analyzing RNase R-treated data. Using RiboMinus/RNase R-treated samples and the corresponding RiboMinus data as the control set, we employed statistical models to fit RNase R treatment efficiency. After correcting RNase R treatment efficiency, we obtained more reliable expression levels for circRNAs detected in these samples. However, for samples without control set, an empirical distribution for RNase R treatment coefficient can be used for posterior distribution. Accurate quantification of circRNAs is the foundation for differential expression analysis. For many studies without replicates, CIRIquant can assign reliable statistics for expression change based on posterior distribution of log fold change. The DE score provides reliable rankings for differentially expressed circRNAs by considering both fold change and p-value. In RNase R-treated samples, CIRIquant can correct DE score based on the posterior distribution of RNase R efficiency, and filter out the majority of circRNAs with relative low p-value and

log fold change. For studies with replicates, CIRIquant adapts the statistical models in edgeR[29] to identify differentially expressed circRNAs. To remove systematic technical effects of library size, a normalization step is performed using TMM normalization factors determined from gene expression levels, then the generalized linear model is used to estimate statistical significance of circRNA expression changes.

Accurate determination of the junction ratio for circRNAs will help us understand the mechanism of circRNA biogenesis and prioritize functionally significant circular transcripts. With the reconstructed circular reference sequences, CIRIquant can transform the reverse and split mapping of junction reads to the direct and continuous alignment of RNA-seq reads against circular reference, which can maximally improve the accuracy of circRNA expression quantification. Extensive evaluations on both simulated and real datasets, as well as experimental validations, demonstrated that CIRIquant can accurately determine the junction ratio of circRNAs and identify linear-circular isoform switching events from paired samples. Notably, CIRIquant embedded a convenient pipeline for differential expression analysis. Using junction ratio as a new feature, different pattern of junction ratio change was observed in three splicing factor knockdown datasets, which provides novel insights on how splicing factors affect circRNA biogenesis. We further applied CIRIquant to 20 paired tumor-normal samples from HCC patients, and identified a number of DE mRNAs, DE circRNAs and DS circRNAs. Notably, there are very few overlapped genes among these three groups, and each group exhibit distinct expression patterns. Moreover, we identified LC-switching and CTU-switching circRNAs, which had undergone significant switching events of junction ratio and CTU. Taken together, we develop a comprehensive method to quantify circRNAs with functional potential and provide a new angle for circRNA prioritization.

## Methods

**Detection and quantification of junction reads.** CIRIquant requires a configuration file in YAML format, which contains the reference sequence in FASTA format and its annotation in GTF format, along with BWA and HISAT2[23] index of the reference sequence for read alignment. In the first step, CIRIquant filters out reads with strong evidence of spliced alignment (minimum mapped segment length >5 bp), and the unmapped reads are used as candidates for circRNA detection. In default, CIRIquant uses CIRI2[12] for circRNA detection. However, a manual input of back-spliced junction sites in BED format generated by any other detection tools is also supported. A pseudo reference of circular sequence is generated by repeating whole length of back-spliced region twice. Then, all candidate reads are aligned to the pseudo-reference sequence using HISAT2, where read pairs that are mapped concordantly across 10 bp region of the junction site are considered as circular reads. Moreover, non-circular reads aligned across the back-spliced junction sites are used to calculate circRNA junction ratio.

**Correction of RNase R treatment coefficient.** For RNase R-treated sample, CIRIquant employs a Gaussian mixture model to fit the enrichment coefficient distribution. The enrichment coefficient is calculated using the following equation:

$$\text{Eff}(\text{Enrichment}) = \frac{\text{BSJ}_{RNase\ R}}{\text{FSJ}_{RNase\ R}} \div \frac{\text{BSJ}_{RiboMinus}}{\text{FSJ}_{RiboMinus}} \tag{1}$$

For all circRNAs that can be detected in both RiboMinus and RiboMinus/RNase R-treated samples, the distribution of enrichment coefficient is fitted using the expectation-maximization algorithm in the Gaussian Mixture Model (GMM) from scikit-learn[48]. The Bayesian information criterion (BIC) criterion is used for selection of components number in GMM. The number of FSJ reads in the ribo-Minus data from the corresponding junction site is extracted from the alignment results, and the corrected number of BSJ reads is calculated based on the number of FSJ reads multiplied by junction ratio estimated by GMM. Subsequently, the mean value of main component in GMM is used as regression coefficient to estimate corrected expression values. The fitted model is then used as prior distribution of RNase R enrichment coefficient, and the posterior distribution of corrected circRNA expression values is used for differential expression analysis.

Then, the fitted distribution of RNase R treatment efficiency is used for estimation of number of circRNA expression values before RNase R treatment.

**DE-score and DS-score calculation**. For RNA-seq data without replicates, the generalized fold change[32] of observed junction reads number is used as a measure of circRNA expression change. In brief, the BSJ read count of a gene can be modeled by the Poisson distribution, so the probability of observing k BSJ reads from a circRNA can be defined as:

$$P(k) = \frac{\lambda^k e^{-\lambda}}{k!}, \lambda = n \times x \tag{2}$$

where λ is the expression level of circRNA, measured by CPM (Count Per Million mapped reads) and n is a normalization factor of sequencing depth. In the Bayesian view, the posterior distribution of λ is defined by:

$$\text{Post}(\lambda) \propto \frac{\lambda^k e^{-k}}{k!} \tag{3}$$

which is a gamma distribution with shape $k+1$ and scale 1. Thus, under two different conditions, the posterior distribution of expression levels $x_1$ and $x_2$ and the log2 fold change $z = \log_2(x_2/x_1)$ can be effectively calculated. Then, the DE-score can be calculated by generalized fold change, utilizing the variance information of the posterior distribution of fold change:

$$DE-score = \begin{cases} \max(t,0) | P_z(z \le t) = 0.05, \text{ if mean}(z) \ge 0 \\ \min(t,0) | P_z(z \ge t) = 0.05, \text{ if mean}(z) < 0 \end{cases} \tag{4}$$

The output DE score balances the fold change and p-value methods, which provides an effective way to ranking differentially expressed circRNAs.

Similarly, DS-score can be calculated in a similar way. For a circRNA with c BSJ reads and 1 FSJ reads, the junction ratio r can be modeled by the beta distribution with shape of c and 1:

$$\text{Post}(r) = \text{Beta}(c, l) \tag{5}$$

then, the DS-score of log2 fold change $y = \log_2(r_2/r_1)$ can be calculated using the posterior distribution of junction ratio under two conditions:

$$DS-score = \begin{cases} \max(t,0) | P_r(r \le t) = 0.05, \text{ if mean}(y) \ge 0 \\ \min(t,0) | P_r(r \ge t) = 0.05, \text{ if mean}(y) < 0 \end{cases} \tag{6}$$

**Differential expression analysis of circRNAs**. For samples with no biological replicate, the Fisher's exact test was performed on a $2 \times 2$ table of BSJ reads of circRNA and total mapped reads, in order to determine whether there is a significant difference between expression level of circRNAs in two different conditions. Furthermore, the rate-ratio.test package in R was used for estimation of change in circRNA junction ratio.

For RNA-seq datasets with biological replicates, a modified version of edgeR[29] is implemented in CIRIquant for differential expression analysis. Briefly, sample specific effect is removed by computing normalization factors using trimmed mean of M-values (TMM) to minimize the log-fold change between samples for most genes. These factors are used for circRNA abundance normalization. Next, the default protocol of generalized linear models (GLM) is performed to estimate the dispersion rate and likelihood ratio test for difference between two groups of samples.

**Simulated data**. We used CIRI-simulator[11] to generate simulated RNA-seq datasets. Human reference genome sequence and its gene annotation GTF file from GENCODE Release 19 (GRCh37.p13) were used for simulated dataset generation. The read length was set to 100 bp, which can assure a balanced performance for the tested methods. Insert length was simulated as the mixture of two normal distribution N(320,70) and N(550,70). Exonic circRNAs were generated in the final simulated datasets using empirical expression distributions, and the coverage of linear transcripts was set to 10-fold.

To evaluate the performance of CIRIquant and other computational tools, we performed CIRI2, CIRCexplorer2, find_circ and KNIFE separately to identify putative back-spliced junction sites, and then used CIRIquant to quantify circRNA expression. For each tool, we computed the Pearson correlation coefficient between its reported number of BSJ reads and the actual value. Considering that the majority of circRNAs were detected at low-expression levels, raw read counts without transformation were used for comparison. For Sailfish-cir, which cannot directly give the number of junction reads, the BSJ reads number can be calculated as follow:

$$\text{BSJ read count} = \text{Num reads} / \text{Effective length} \tag{7}$$

where the num reads and effective length can be directed output from Sailfish-cir results. Then, the Pearson correlation coefficient was used for performance measurement.

To assess the performance of RNase R treatment efficiency correction, we calculated the deviation ratio between the observed expression values before and after correction in RiboMinus/RNase R-treated library and the expected value in

the RiboMinus sample:

$$\text{deviation ratio} = \frac{\text{CPM}_{\text{RiboMinuss/RNase }R} - \text{CPM}_{\text{RiboMinus}}}{0.5 * \left( \text{CPM}_{\text{RiboMinusRiboMinus/RNase RRNase }R} + \text{CPM}_{\text{RiboMinus}} \right)} \tag{8}$$

where the deviation ratio represents the bias caused by RNase R treatment in expression level estimation. The deviation ratio at zero indicates that the expression level of circRNA remain the same under two different conditions.

**Cell culture and knockdown assay**. Human HeLa cells were obtained from the American Type Culture Collection (ATCC CCL-2) and expanded in Dulbecco's Modified Eagle Medium (DMEM) supplemented with 10% fetal bovine serum (FBS) and 1% penicillin-streptomycin (Gibco, USA). The lentiviral vectors carrying TRA2B, PTBP1, and MBNL1 shRNAs were constructed based on sequence AGCTAAAGAACGTGCCAAT, GCACAGTGTTGAAGATCAT, and GCCTGCTTTGATTCATTGAAA, respectively. An adenoviral vector that carried pSIH-H1-copGFP was purchased from System Biosciences (Mountain View, CA, USA). HEK 293 T cells were transfected with shRNA plasmids and psPAX2, pMD2.G at a ratio of 4:3:1. Then, virus was collected after 48 h, and the infections were conducted for 48 h. HeLa cells were selected with 2 μg/ml puromycin after transduction.

**RNA isolation, library preparation and RNA-seq**. Following RNA isolation by TRIZOL (Invitrogen, Carlsbad, CA), the quality and quantity of RNA were assessed with NanoDrop, Qubit and BioAnalyzer 2100 (Agilent). A RiboMinus kit (KAPA, USA) was used for ribosomal RNA depletion. Then, total RNA was divided into three equal parts after depletion of ribosomal RNA to construct a riboMinus and two riboMinus/RNase R-treated replicates libraries. RNase R was purchased from Epicentre (Madison, WI) to digest linear transcripts. rRNA-depleted RNA was incubated at 37 °C with 10 U/μg RNase R in 16 μl reaction.

For each cell line with shRNA-mediated knockdown of splicing factors, a RiboMinus and two RiboMinus/RNase R-treated replicates libraries were constructed. Both libraries were prepared following the TruSeq protocol (Illumina, San Diego, CA). Illumina HiSeq 2500 platform at the Research Facility Center at Beijing Institutes of Life Science, CAS was used for sequencing with $2 \times 101$ bp paired reads.

**Experimental validation**. To verify the expression level and junction ratio of predicted circRNA transcripts, outward primers set (Supplementary Table 5) were designed to quantify the expression of five circRNAs in control and MBNL1/PTBP1/TRA2B-knockdown samples. Specifically, additional primers targeting the 5′ and 3′ cirexon of these circRNAs were also used for quantification of forward splice junction reads. The relative expression of RNA was calculated using GAPDH as a control. For hsa_intergenic_006404, the sequence of PCR products was determined using Sanger sequencing.

**Public RNA-seq data**. Public RNA-seq data from seven circRNA studies were downloaded from the NCBI SRA database, including four species, *Caenorhabditis elegans* (SRP050505), *Drosophila Melanogaster* (PRJNA241181), *Mus Musculus* (PRJNA294035) and *Homo Sapiens* (SRP011042, SRP052817, SRP067050). The reference genomes of human (Release 19, GRCh37.p13) and mouse (Release M9, GRCm38.p4) were downloaded from GENCODE. The reference genomes of *Caenorhabditis elegans* (WBcel235) and *Drosophila Melanogaster* (BDGP6) were downloaded from Ensembl. Ribosomal RNA sequences for all four species were downloaded from NCBI Nucleotide database. Paired samples of RNase R-treated and untreated libraries were used for the measurement of circRNA enrichment coefficient. RiboMinus treated RNA-seq datasets of tumor (SRX1558046-SRX1558064) and normal liver samples (SRX1558026-SRX1558045) from 20 HCC patients generated in a previous study[40] were used for differential expression analysis.

Raw reads were first aligned to rRNA index using HISAT2, where mapped reads were removed, and the mapping rate was calculated as a measurement for rRNA percentage. Then, CIRIquant was employed for circRNA characterization and quantification.

**Gene expression analysis in RNA-Seq data**. For gene expression analysis, analysis protocol with HISAT2[23] and StringTie[27] was performed. In brief, the sequencing reads were aligned to the reference genome using HISAT2, and StringTie was used to re-assemble the transcriptome, as well as to quantify gene expression levels. Differential expression analysis was performed using EdgeR[29], with the same parameters as those in circRNA differential expression analysis.

**Splicing factor binding site**. Public CLIP-seq datasets were downloaded from iCLIPdb[36], where the called peak of TRA2B and PTBP1 were used for binding site coverage calculation. For circRNA and alternative skipped mRNA exons, we scanned exon and the upstream and downstream 2 kb region using a sliding window with size of 8 nt.

**Tissue specificity of circRNAs**. The expression pattern of circRNAs in human tissues were downloaded from the circAtlas database (http://circatlas.biols.ac.cn/). RNA-seq datasets from RiboMinus RiboMinus/RNase R-treated samples were used for tissue specificity assessment[43]. The relative expression level was calculated by dividing the CPM of circRNA in each tissue by the sum of expression values in all tissues.

**Reporting summary**. Further information on research design is available in the Nature Research Reporting Summary linked to this article.

## Data availability

The raw sequence data generated in this study have been deposited in the Genome Sequence Archive[49] in BIG Data Center, Beijing Institute of Genomics, Chinese Academy of Sciences (https://bigd.big.ac.cn/gsa), with the accession number CRA001838. Details about data generated in previous studies and analyzed in this study were included in Supplementary Table 1 and the Methods section.

## Code availability

CIRIquant is implemented in Python, which can be freely accessed at https://sourceforge.net/projects/ciri. The software is packaged with test data, and has been extensively tested on Linux and OS X.

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

## Acknowledgements

This work was supported by grants from National Key R&D Program [2018YFC0910400], National Natural Science Foundation of China [31722031, 91640117, 91940306,31671364, 91531306].

## Author contributions

F.Z conceived the project. J. Z implemented the tool and analyzed the data. S. C and J. Y performed the experiments and generated RNA-seq data. J.Z. and F.Z. wrote the manuscript.

## Competing interests

The authors declare no competing interests.
