## [Peer Review File · Nature Communications]

Reviewers' comments:

Reviewer #1 (Remarks to the Author):

Zhang et al. proposed a new algorithm, CIRIquant, to quantify circular RNA from RNA-seq data. By employing pseudo-circular reference re-alignment and RNase R treatment correction, CIRIquant provides more accurate quantification and differential analysis of circular RNAs over other existing methods. The authors showed the potential of CIRIquant by applying to liver cancer samples. This study provides the community with an efficient and useful tool for circular RNA analysis.

Major comments:

1. Are the 34 pairs of RiboMinus and RiboMinus/RNase R samples (should be 68 samples?) included in the collected 64 samples? In addition, Table S1 does not seem to include all the collected samples. It would be better to clarify these sample numbers more clearly.
2. How much could the pseudo-reference re-alignment strategy reduce the false positive BSJs? It would be better to compare the BSJs before and after pseudo-reference re-alignment.
3. For the LC-switching circular RNA, hsa_intergenic_006404, the authors observed enrichment of H3K27ac. This is very interesting. Is this circular RNA located within or near known functional genomic regions? For example, it may be near regulating lncRNAs or enhancer RNAs. The author may further examine this.
4. The last section for studies in hepatocellular carcinoma is extremely interesting. Is that possible for the authors to run a similar analysis in an independent dataset to confirm their observations, in particular for LC/CTU-switching?

Minor comments:

1. For Figure 1B, could the authors label the sample information clearly on Y-axis?
2. The full name of "FSJ" should be clarified.
3. Test data is not packaged in the software (download 09/09/2019). It would be better to include test data.

Reviewer #2 (Remarks to the Author):

In this manuscript, the authors proposed a computational method, CIRIquant, to quantify circular RNA (circRNA) expression from both ribo-depleted and RNase R-treated RNA-seq data. Basically, CIRIquant quantifies circRNA expression by estimating the number of back-splicing junction (BSJ) reads through re-aligning sequencing reads against pseudo-circular reference. Compared with the other algorithms that are based on counting BSJ reads number (e.g. KNIFE, CIRCexplorer2, DCC, etc), CIRIquant apparently has a better BSJ quantification accuracy. Also, by employing sophisticated statistical methods, CIRIquant is able to correct the bias caused by RNase R treatment and thus improve the quantification performance upon RNase R-treated RNA-seq data. Generally, the manuscript is well-written and informative.

Major concerns

- One of my major concerns is regarding the comparison between CIRIquant and Sailfish-cir. Sailfish-cir is a model-based circRNA quantification tool, which was designed to directly estimate circular transcript expression, measured as TPM (transcripts per million reads), instead of counting the number of BSJ reads. In this manuscript, to compare the quantification performance between CIRIquant and Sailfish-cir, the BSJ reads number for Sailfish-cir was imputed as number of circular reads / effective length. I agree that both the number of circular reads and effective length are the outputs of Sailfish-cir. However, I don't think that the division quotient between number of circular reads and effective length can accurately reflect the number of BSJ reads. As we know, RNA-seq reads are not evenly distributed along both linear and circular RNA transcripts. Therefore, it is unfair to simply compare the BSJ reads number generated from CIRIquant against the division quotient computed from Sailfish-cir, i.e. dividing number of circular reads by effective length. As I mentioned above, Sailfish-cir was not designed to compute BSJ reads number. For a fair

comparison, I would suggest the authors, if doable, to transform the BSJ reads count generated by CIRIquant into transcript TPM format.

- My second major concern is regarding the qRT-PCR validation. The authors performed qRT-PCR validation for five randomly selected circRNAs. However, I didn't see the comparison of CIRIquant against the other computational tools regarding the qRT-PCR data. I am curious whether CIRIquant is the one that best fits the qRT-PCR data points, compared with KNIFE, CIRCexplorer2, Sailfish-cir, etc.

- I would suggest the authors to perform a comparison of CIRIquant against the other circRNA quantification tools on a third-party dataset, in which both RNA-seq and qRT-PCR data are available, e.g. Szabo et al. *Genome Biol.* 2015 (doi: 10.1186/s13059-015-0690-5)

Minor concerns

- In the Introduction section, the authors mentioned that "only CIRI2, DCC14 and Sailfish-cir can output the junction ratio". Actually, Sailfish-cir directly outputs circRNA TPM and reads count, instead of junction reads ratio.

- Please provide the details describing how the circRNA coverage was calculated in Figure 3B.

- On page 7, the authors claimed that, based on qRT-PCR data, "only CIRIquant can achieve unbiased quantification for all types of circRNAs." However, I didn't see a comparison of CIRIquant against the other computational tools in Supplementary Figure S3.

- Typo: please rephrase "five randomly select circRNAs" as "five randomly selected circRNAs" on page 7.

Reviewers' comments:

Reviewer #1 (Remarks to the Author):

Zhang et al. proposed a new algorithm, CIRIquant, to quantify circular RNA from RNA-seq data. By employing pseudo-circular reference re-alignment and RNase R treatment correction, CIRIquant provides more accurate quantification and differential analysis of circular RNAs over other existing methods. The authors showed the potential of CIRIquant by applying to liver cancer samples. This study provides the community with an efficient and useful tool for circular RNA analysis.

Responses: We greatly appreciate the reviewer's comments on the novelty and significance of our approach. In this revised version, we extensively revised the manuscript and added more analyses. Please refer to the following responses for detail.

Major comments:

1. Are the 34 pairs of RiboMinus and RiboMinus/RNase R samples (should be 68 samples?) included in the collected 64 samples? In addition, Table S1 does not seem to include all the collected samples. It would be better to clarify these sample numbers more clearly.

Response: We thank the reviewer for pointing this out. In this study, we initially constructed four cell lines with shRNA-mediated knockdown of splicing factors (SRSF3 / PTBP1 / TRA2B / MBNL1). For these four knockdown samples and a mock treatment control, the total RNA was extracted and a RiboMinus RNA-seq library and two RiboMinus/RNase R treated replicates libraries were constructed, which made up of two pairs of RNase R treatment samples. In addition, 48 public RNA-seq datasets consisting of 24 pairs of RiboMinus/RNase R samples were also used in Figure 1 A-D, which were listed in **Table S1**. Therefore, the total amount of RiboMinus and RiboMinus/RNase R pairs was $24 + 5 \times 2 = 34$ and the total number of collected samples was $48 + 5 \times 3 = 63$. RNA-seq datasets generated in our study have been submitted to the Genome Sequence Archive in BIG Data Center (<https://bigd.big.ac.cn/gsa>), with the accession number CRA001838.

2. How much could the pseudo-reference re-alignment strategy reduce the false positive BSJs? It would be better to compare the BSJs before and after pseudo-reference re-alignment.

Responses: As suggested, we compared the false positive rate of BSJ identification before and after pseudo reference re-alignment using the simulated dataset in **Figure S4**. Firstly, we ran all five tools to detect circRNAs, and then applied CIRIquant using the predicted junction sites as input respectively. Considering that most tools cannot output the ID of BSJ reads, we chose the false discovery rate of circRNA detection for performance assessment. As shown in **Figure S2**, after pseudo reference re-alignment, CIRIquant can efficiently reduce the false discovery rate and improve the Pearson correlation coefficient for all these methods.

3. For the LC-switching circular RNA, hsa_intergenic_006404, the authors observed enrichment of H3K27ac. This is very interesting. Is this circular RNA located within or near known functional genomic regions? For example, it may be near regulating lncRNAs or enhancer RNAs. The author may further examine this.

Responses: As suggested, we explored the functional genomic regions near hsa_intergenic_006404 using the UCSC genome browser (**Figure S10**). We found that this circRNA is derived from an enhancer region, and it's overlapped with DNase I hypersensitivity peak and transcription factor CHIP-seq clusters.

4. The last section for studies in hepatocellular carcinoma is extremely interesting. Is that possible for the authors to run a similar analysis in an independent dataset to confirm their observations, in particular for LC/CTU-switching?

Responses: As suggested, we downloaded public RNA-seq datasets from two previous studies (PRJNA232669 and PRJNA428447). The first dataset contains 7 normal human brain samples and 8 advanced Alzheimer's disease brain samples; the second dataset contains 7 pairs of carcinoma/normal tissues from patients with renal cell carcinoma. For both datasets, we implemented CIRIquant for circRNA quantification, and characterized LC-switching and CTU-switching events detected in these samples. As shown in **Figure S9**, we observed a number of LC-switching and CTU-switching circRNAs in both the Alzheimer's disease and Renal cell carcinoma samples, which further confirmed our findings on human liver cancer. Please refer to the **Figure S9** for details.

Minor comments:

1. For Figure 1B, could the authors label the sample information clearly on Y-axis?

Responses: As suggested, we labeled the sample name of Y-axis in **Figure 1B**.

2. The full name of "FSJ" should be clarified.

Responses: As suggested, we clarified the full name of forward-spliced junction (FSJ) reads in the main text.

3. Test data is not packaged in the software (download 09/09/2019). It would be better to include test data.

Response: Now we include the test data in our SourceForge project (https://sourceforge.net/projects/ciri/files/CIRIquant/test_data.tar.gz/download), including the demo for both quantification and differential expression analysis.

Reviewer #2 (Remarks to the Author):

In this manuscript, the authors proposed a computational method, CIRIquant, to quantify circular RNA (circRNA) expression from both ribo-depleted and RNase R-treated RNA-seq data. Basically, CIRIquant quantifies circRNA expression by estimating the number of back-splicing junction (BSJ) reads through re-aligning sequencing reads against pseudo-circular reference. Compared with the other algorithms that are based on counting BSJ reads number (e.g. KNIFE, CIRCexplorer2,

DCC, etc), CIRIquant apparently has a better BSJ quantification accuracy. Also, by employing sophisticated statistical methods, CIRIquant is able to correct the bias caused by RNase R treatment and thus improve the quantification performance upon RNase R-treated RNA-seq data. Generally, the manuscript is well-written and informative.

Response: We greatly appreciate the reviewer's comments on our approach. In this revised manuscript, we included more comparison suggested by the reviewer. Please refer to the following responses for detail.

Major concerns

- One of my major concerns is regarding the comparison between CIRIquant and Sailfish-cir. Sailfish-cir is a model-based circRNA quantification tool, which was designed to directly estimate circular transcript expression, measured as TPM (transcripts per million reads), instead of counting the number of BSJ reads. In this manuscript, to compare the quantification performance between CIRIquant and Sailfish-cir, the BSJ reads number for Sailfish-cir was imputed as number of circular reads / effective length. I agree that both the number of circular reads and effective length are the outputs of Sailfish-cir. However, I don't think that the division quotient between number of circular reads and effective length can accurately reflect the number of BSJ reads. As we know, RNA-seq reads are not evenly distributed along both linear and circular RNA transcripts. Therefore, it is unfair to simply compare the BSJ reads number generated from CIRIquant against the division quotient computed from Sailfish-cir, i.e. dividing number of circular reads by effective length. As I mentioned above, Sailfish-cir was not designed to compute BSJ reads number. For a fair comparison, I would suggest the authors, if doable, to transform the BSJ reads count generated by CIRIquant into transcript TPM format.

Response: We agree with the reviewer that simply use the ratio of NumReads / Effective Length from Sailfish-cir results may not be appropriate. As suggested, we employed Transcripts Per Million (TPM) to compare the performance between CIRIquant and Sailfish-cir in revised version of **Figure 3B**. According to the definition of TPM (Wagner et al, Theory in Biosciences, 2012), the TPM of circRNA i can be calculated as

$$TPM_i = \frac{r_i \times rl \times 10^6}{fl_i \times T} \quad (1)$$

$$T = \sum_{i \in Gene + Circ} \frac{r_i \times rl}{fl_i} \quad (2)$$

where r_i is the read count of circRNA i , rl is the sequencing read length, and fl_i is circRNA i 's real length. In our simulation, we firstly randomly assigned coverage cov_i to circRNA i , then simulated reads in a uniform distribution on this circular

transcript, with the total reads number as $cov_i \times fl_i$. Thus, according to equation (1), the real TPM of circRNAs should be linearly correlated to their simulated coverage:

$$TPM_i = \frac{rl \times 10^6}{T} \times cov_i$$

Therefore, the Pearson correlation coefficient (r) was used to evaluate the linear relationship between TPM by Sailfish-cir. As shown in **Figure 3B**, CIRIquant (r =0.947) outperformed Sailfish-cir (r =0.618) in the simulation results, which is consistent with our previous finding.

- My second major concern is regarding the qRT-PCR validation. The authors performed qRT-PCR validation for five randomly selected circRNAs. However, I didn't see the comparison of CIRIquant against the other computational tools regarding the qRT-PCR data. I am curious whether CIRIquant is the one that best fits the qRT-PCR data points, compared with KNIFE, CIRCexplorer2, Sailfish-cir, etc.

Responses: As suggested, we have updated **Figure S4** to include the comparison of different tools. For the randomly selected circRNAs, we used the qPCR results to calculate the relative expression level compared to GAPDH in four RiboMinus samples without RNase R treatment, then applied all six methods to RiboMinus/RNase R treated samples. For CIRIquant, we first compared the raw results using RiboMinus/RNase R treated samples only, and then performed RNase R correction using RiboMinus data as control and estimated the corrected expression level of these circRNAs in the RiboMinus data. We calculated the root-mean-squared error (RMSE) of the real qRT-PCR results and the predicted expression level derived from each tool. As shown in **Figure S4**, after the correction of RNase R treatment efficiency, CIRIquant can effectively reflect the real expression level before RNase R treatment, while all other methods exhibited poor performance on quantifying circRNAs from RNase R treated samples.

- I would suggest the authors to perform a comparison of CIRIquant against the other circRNA quantification tools on a third-party dataset, in which both RNA-seq and qRT-PCR data are available, e.g. Szabo et al. Genome Biol. 2015 (doi: 10.1186/s13059-015-0690-5)

Responses: We greatly appreciate the reviewer for this suggestion. We downloaded the RNA-seq and qRT-PCR data from Szabo et al's study, and then evaluated the performance of different tools on this dataset. To gain a fair comparison, we used TPM for Sailfish-cir and CPM (counts per million, calculated as #BSJ / mapped reads) for all the other BSJ-based tools to normalize the effect of data size in different samples. In order to show the performance of circRNA quantification, we fitted the log2 average expression level determined by each tool to qRT-PCR's CT values provided by Szabo et al's study, and then calculated the correlation coefficient (r) and p values of the regression model. As shown in **Figure S3**, compared with other tools, CIRIquant obtained the best result regarding both the correlation coefficient (r = -0.79)

and p value ($p = 3.8 \times 10^{-18}$).

Minor concerns

- In the Introduction section, the authors mentioned that “only CIRI2, DCC14 and Sailfish-cir can output the junction ratio”. Actually, Sailfish-cir directly outputs circRNA TPM and reads count, instead of junction reads ratio.

Responses: Thanks for pointing this out. It has been revised accordingly.

- Please provide the details describing how the circRNA coverage was calculated in Figure 3B.

Responses: The simulated coverage of circRNA has been described above. As the reviewer mentioned, the coverage from Sailfish-cir was calculated using the division quotient of NumReads and Effective Length, which may be unfair for a direct comparison. Thus, we used TPM of the identified circRNAs from Sailfish-cir for comparison (**Figure 3B**).

- On page 7, the authors claimed that, based on qRT-PCR data, “only CIRIquant can achieve unbiased quantification for all types of circRNAs.” However, I didn’t see a comparison of CIRIquant against the other computational tools in Supplementary Figure S3.

Responses: As suggested, we compared the performance of all the tools on the dataset from Szabo *et al. Genome Biol. 2015*. Please refer to our previous responses and **Figure S3** for details.

- Typo: please rephrase “five randomly select circRNAs” as “five randomly selected circRNAs” on page 7.

Responses: Thanks. It has been revised accordingly.

REVIEWERS' COMMENTS:

Reviewer #1 (Remarks to the Author):

The authors addressed all of my concerns.

Reviewer #2 (Remarks to the Author):

The authors have addressed all my comments.